# The Impact of Liver Steatosis on Interleukin and Growth Factors Kinetics during Chronic Hepatitis C Treatment

**DOI:** 10.3390/jcm13164849

**Published:** 2024-08-16

**Authors:** Leona Radmanic Matotek, Snjezana Zidovec-Lepej, Nikolina Salek, Adriana Vince, Neven Papic

**Affiliations:** 1Department of Immunological and Molecular Diagnostics, University Hospital for Infectious Diseases “Dr. Fran Mihaljević”, 10000 Zagreb, Croatia; lradmanic@bfm.hr (L.R.M.); szidovec@bfm.hr (S.Z.-L.); 2Department for Viral Hepatitis, University Hospital for Infectious Diseases “Dr. Fran Mihaljević”, 10000 Zagreb, Croatia; njurisic@bfm.hr (N.S.); avince@bfm.hr (A.V.); 3Department for Infectious Diseases, School of Medicine, University of Zagreb, 10000 Zagreb, Croatia

**Keywords:** chronic hepatitis C, HCV, liver steatosis, cytokines, growth factor, EGF, VEGF, fibrosis

## Abstract

**Background/Objectives:** Various biological response modifiers play important roles in the immunopathogenesis of chronic hepatitis C (CHC). While serum levels of cytokines and growth factors change with the disease severity and treatment responses, the impact of concomitant liver steatosis on systemic inflammatory response is largely unknown. The aim of this study was to analyze the characteristics and kinetics of serum profiles of interleukins and growth factors in CHC patients with steatotic liver disease (SLD). **Methods:** Serum concentrations of 12 cytokines (IL-5, IL-13, IL-2, IL-6, IL-9, IL-10, IFN-γ, TNF-α, IL-17A, IL-17F, IL-4 and IL-22) and 6 growth factors (Angiopoietin-2, EGF, EPO, HGF, SCF, VEGF) were analyzed in 56 CHC patients at four time points (baseline, week 4, week 8 and SVR12) with bead-based flow cytometry assay. **Results:** At baseline, patients with SLD had significantly lower IL-9, IL-10, IL-13 and IL-22 and higher serum concentrations of EGF, VEGF and ANG. In a subgroup of patients with advanced liver fibrosis, SLD was linked with lower serum concentrations of IL-4, IL-5, IL-9, IL-10, IL-13 and IL-22 and higher concentrations of HGH and VEGF. Distinct cytokine kinetics during DAA treatment was observed, and SLD was identified as the main source of variation for IL-5, IL-9, IL-10, IL-13, IL-17A, IL-22, EGF, VEGF and ANG. Patients with SLD at SVR12 had significantly higher VEGF and HGF serum concentrations. **Conclusions:** SLD is associated with distinct cytokine and growth factor profiles and kinetics during CHC treatment, which might be associated with disease severity and the capacity for liver regeneration and contribute to fibrosis persistence.

## 1. Introduction

Hepatitis C virus (HCV) is a hepatotropic RNA virus causing chronic infection in up to 85% of infected patients, with an estimated 50 million people living with chronic hepatitis C (CHC) in 2024 [1]. HCV productively infects hepatocytes to impart liver inflammation and progressive tissue damage [2]. This complex virus–host interaction leads to the activation of both parenchymal and non-parenchymal liver cells and changes in the expression of various biological response modifiers, including interleukins, cytokines and growth factors, which drive liver inflammation and lead the process of fibrogenesis to the development of cirrhosis and, potentially, hepatocellular carcinoma (HCC) [3,4,5]. The elucidation of these complex molecular networks is crucial for understanding CHC immunopathogenesis. Serum levels of several cytokines and growth factors change with the states of infection and treatment and may serve as potential biomarkers for disease progression and therapeutic effects [4,6,7]. While the HCV treatment with directly acting antiviral drugs (DAA) enables cure in more than 95% of infected individuals, the currently available literature does not fully explain the mechanisms of inflammation and fibrosis reversibility after HCV eradication.

Furthermore, systemic inflammatory milieu might be influenced by multiple patients’ factors, including comorbidities, metabolic syndrome, obesity, chronic medications, etc. Chronic hepatitis C is frequently accompanied by liver steatosis in 50–80% of patients, a 2.5-fold higher prevalence than in the general population [8]. Steatotic liver disease (SLD) is the most common liver disease in Western countries, being closely linked with components of metabolic syndrome, primarily insulin resistance, obesity and hyperlipidemia [9]. However, the molecular mechanisms of steatosis in the setting of HCV infection are more complex and, in addition to the usual metabolic factors, include HCV-induced alteration of cellular lipid metabolism, which is specifically linked to genotype 3 infection [2,10].

The association of liver steatosis with CHC disease progression is still a matter of debate; while some studies have shown a positive correlation between fibrosis progression and development of HCC, other reported no correlation [11,12,13]. Steatosis persisted or was even newly developed in a significant number of patients after achieving SVR [14,15,16]. Notably, SLD is associated with chronic low-level inflammation, vascular dysfunction, impaired insulin sensitivity, lipotoxicity and compromised systemic immune responses, all of which might play an important role in CHC pathogenesis [17]. However, reports on the impact of concomitant liver steatosis in CHC on systemic inflammatory responses before and after DAA therapy are scarce. While SLD persists in a significant number of patients even after SVR, its impact on cytokine profiles after virus elimination is unknown. Therefore, there is a significant gap in our understanding of HCV treatment’s impact on the systemic inflammatory milieu in this patient group.

The aim of this study was to analyze the characteristics and kinetics of interleukins and growth factor serum profiles in patients with CHC and liver steatosis during DAA treatment and its association with the persistence of steatosis after viral clearance.

## 2. Materials and Methods

### 2.1. Study Design and Patients

This monocentric study was conducted at the University Hospital for Infectious Diseases Zagreb (UHID), Croatia, and was part of a prospective longitudinal cohort study that recruited patients with chronic hepatitis C who were evaluated for DAA treatment. In this study, 56 patients with CHC (≥18 years old) were included for the analysis of interleukin and growth factor serum kinetics during DAA treatment. All patients achieved a SVR, defined as a negative HCV RNA PCR test (Abbott Molecular, Des Plaines, IL, USA) 12 weeks after completing the treatment (SVR12). Excluded were patients with known malignant, autoimmune or hematological diseases, as well as those with significant alcohol consumption. All patients were HBV- and HIV-negative. The protocols for sample collection and investigation were approved by the UHID Ethics Committee (code 01-673-1-2021), and all patients signed an informed consent form.

### 2.2. Data Collection and Definitions

Routine clinical data, including demographics, comorbidities, chronic medication use, clinical status, anthropometric measurements, and routine laboratory tests, were collected.

The degree of liver steatosis was assessed using the controlled attenuation parameter (CAP), a method for grading steatosis by measuring the degree of ultrasound attenuation by hepatic fat (FibroScan, Echosens, France). Fibrosis stages were classified according to METAVIR score as F0/F1 < 7.0 kPa, F2 7.0–9.5 kPa, F3 < 9.5 kPa and F4 > 12.5 kPa [18]. Patients were diagnosed with liver steatosis if they had CAP > 238 dB/m and further classified into three groups: grade I CAP 238–260 dB/m (S1 ≥ 10% of hepatocytes with fat accumulation), grade II CAP 260–299 (S2 ≥ 33% of hepatocytes with fat accumulation) and grade III CAP > 300 dB/m (S2 ≥ 66% of hepatocytes with fat accumulation) [18]. At least 10 valid measurements per participant were obtained using the M or XL probe, as suggested by the manufacturer, and mean liver stiffness measurement (LSM) and CAP with corresponding IQRs were recorded. All patients were required to have a LSM of an at least 60% success rate with an interquartile range of <30% of the median value. In addition, APRI, FIB-4, NAFLD and FAST scores were calculated as markers of liver injury [19,20,21,22].

### 2.3. Sample Collection and Bead-Based Cytometry

Cytokine and growth factor concentrations in patients with CHC were determined before, during and after DAA treatment. The samples were stored at −80 °C in aliquots to avoid repeated freeze and thaw cycles. Serum concentrations of 12 cytokines (interleukin 5 (IL-5), IL-13, IL-2, IL-6, IL-9, IL-10, interferon-gamma (IFN-γ), tumor necrosis factor-alpha (TNF-α), IL-17A, IL-17F, IL-4 and IL-22) and the 6 growth factors angiopoietin-2 (Ang-2), epidermal growth factor (EGF), erythropoietin (EPO), hepatocyte growth factor (HGF), stem cell factor (SCF) and vascular endothelial growth factor (VEGF) were analyzed using bead-based flow cytometry. HumanThCytokine Panel with Filter Plate (Biolegend, San Diego, CA, USA) was used to determine cytokine concentration, and Human Growth Factor Panel with Filter Rate (Biolegend, San Diego, CA, USA) was used to determine growth factor levels according to manufacturer instructions. A flow cytometer BD FACSCanto II (Beckton Dickinson, Franklin Lakes, NJ, USA) was used for detection, and the concentrations of analyzed cytokines and growth factors were analyzed in the LEGENDplex software (version 8.0., Biolegend, San Diego, CA, USA).

### 2.4. Statistical Analysis

Clinical characteristic, laboratory and demographic data were evaluated and presented deceptively. The Shapiro–Wilk test was used to check if a continuous variable followed a normal distribution. The cohort was divided into two groups based on the presence of steatosis, and Fisher’s exact test and the Mann–Whitney U test were used for variables comparison. A repeated measures two-way ANOVA test with Tukey’s multiple comparisons test was used to analyze the kinetics of cytokines and growth factors between two groups (liver steatosis vs. non-steatosis) at 4 time points (baseline, week 4, week 8 and SVR12). Repeated measures three-way ANOVA was used for the analysis of whether any of three variables (liver fibrosis, liver steatosis and time) influences cytokine and growth factor kinetics, as well as the relationship between them. A Wilcoxon matched pairs test was used to analyze the changes in concentrations at baseline and at SVR12. Correlations were analyzed using Spearman’s rank correlation coefficient and summarized in a correlation matrix. All tests were two-tailed; a *p*-value < 0.05 was considered statistically significant. Statistical analyses were performed using GraphPad Prism Software version 10 (San Diego, CA, USA).

## 3. Results

### 3.1. Baseline Patients’ Characteristics

A total of 56 patients with CHC (57.1% males; median age of 55, IQR 46–63 years) were included. Of them, 28 patients had liver steatosis according to CAP findings. Patients with liver steatosis were more frequently obese (25% vs. 3.6%) and more frequently had T2DM, dyslipidemia or arterial hypertension. There were no differences in other comorbidities and chronic medications. The median CAP in patients with liver steatosis was 277 (244–324) dB/m, and in patients without liver steatosis it was 219 (192–224) dB/m. In total, 12 patients had grade 3 steatosis (42.9%), 12 (42.9%) had grade 1 steatosis and 4 (14.3%) had grade 2 steatosis; moreover, 17 (60.71%) patients with SLD and 16 (57.14%) without SLD had advanced liver fibrosis (LS > 9.5 kPa). There were no differences in HCV genotype, viremia and duration of infection and DAA treatment regimen between groups. Except for higher serum fasting glucose and a lower platelet-to-lymphocyte ratio, there were no differences in other routine laboratory findings. The baseline characteristics of CHC patients stratified by the presence of liver steatosis are presented in Table 1.

### 3.2. Baseline Cytokine and Chemokine Concentrations According to Steatosis and Fibrosis Stage

To investigate the impact of liver steatosis on cytokine and growth factor profiles, the serum concentrations of selected biomediators were measured in CHC patients with and without liver steatosis before DAA treatment. As presented in Table 2, most of the measured cytokines had similar concentrations, except for IL-9, IL-10, IL-13 and IL-22, which were significantly lower in patients with steatosis. On the contrary, patients with liver steatosis had significantly higher baseline concentrations of the growth factors EGF, VEGF and ANG (Table 2).

Next, we examined the impact of liver fibrosis in patients with and without steatosis on serum cytokine and growth factor concentrations. A distinct cytokine expression was observed, as is presented in Figure 1. In a group of patients with advanced liver fibrosis, liver steatosis was linked with lower serum concentrations of IL-4, IL-5, IL-9, IL-10, IL-13 and IL-22. In patients with mild liver fibrosis, liver steatosis was associated with lower serum concentrations of IL-13 and IL-17F. There was no impact of liver steatosis on IL-4, IL-5, IL-10 and IL-22 levels in patients with mild fibrosis.

Serum concentrations of EGF were higher in patients with liver steatosis and mild fibrosis, but there was no difference in EGF levels in advanced fibrosis. While in mild fibrosis, steatosis was associated with decreased HGF and VEGF levels, in advanced fibrosis, HGF was increased in patients with steatosis.

Further, we performed a subgroup analysis to analyze if these changes might be related to HCV genotype infection. There were no differences in any of the biomediators measured between genotype 3 and genotype 1 infection.

A correlation analysis was performed to identify potential correlations among paired laboratory parameters, including growth factor and cytokine concentrations and selected clinical variables. CAP, as a measure of liver steatosis, showed positive correlations with EGF (r = 0.33, *p* = 0.0065), VEGF (r = 0.31, *p* = 0.0091) and ANG (r = 0.30, *p* = 0.0116) but a negative correlation with a majority of measured cytokines (e.g., IL-9 r = −0.24, *p* = 0.0399; IL-10 r = −0.21, *p* = 0.05, IL-22 r = 0.34, *p* = 0.006). Liver stiffness had positive correlations with HGF (r = 0.33, *p* = 0.006), IL = 6 (r = 0.3, *p* = 0.012) and IL-17F (r = 0.27, *p* = 0.024). As shown in Figure 2, growth factors showed a negative correlation with measured cytokines, while the correlations between cytokines were mainly positive.

### 3.3. Cytokine and Growth Factor Kinetics in CHC Patients before, during and after DAA Treatment

Next, we examined the impact of liver steatosis on cytokine and growth factor serum kinetics during CHC treatment. We measured concentrations of 12 cytokines and 5 growth factors in sera taken at four different time points (before treatment, after 4 and 8 weeks of DAA therapy and 12 weeks after treatment, e.g., SVR12). Cytokine level comparison at the four selected time points is shown in Figure 3 (and Appendix A).

Distinct cytokine kinetics was observed, and two-way ANOVA analysis identified liver steatosis as source of variation for IL-5, IL-9, IL-10, IL-13, IL-17A and IL-22. Only IL-10 and IL-22 showed variations in serum expression over time. IL-10 concentrations decreased during treatment in patients without liver steatosis but not in patients with liver steatosis. On the contrary, IL-22 concentrations decreased in patients with steatosis but not in those without it. Other measured cytokines exhibited similar concentrations at these four time points, as is presented in Figure 3 (and Appendix A).

EGF exhibited the highest relative increase across the four selected measurement time points, followed by HGF and VEGF. EPO, SCF and ANG showed no variability during this time. The patients with liver steatosis had significantly higher EGF, VGF and ANG serum concentrations at baseline and weeks 4 and 8, while their concentrations at SVR were similar to those of patients without liver steatosis, as shown in Figure 4 (and Appendix A).

We further examined if the cytokine and growth factor serum kinetics in patients with liver steatosis were influenced by the presence of liver fibrosis. In a three-way ANOVA analysis, liver steatosis was linked with the kinetics of EGF, VEGF and ANG, while liver fibrosis was not a source of variation in our analysis (Figure 4). Similarly, as shown in Figure 4, IL-4, IL-10 and IL-13 levels were influenced by the presence of steatosis, while the main source of variation in IL-22 was time.

### 3.4. Association between Cytokine and Growth Factor Concentrations with Steatosis and Fibrosis Regression after DAA Treatment

Liver stiffness decreased in the majority of patients at SVR12. Meanwhile, the median value of CAP did not significantly change: six patients from the SLD group at baseline had no steatosis at SVR12, while five patients from the non-SLD group developed steatosis.

Although there was a significant decrease in VEGF levels in patients with and without liver steatosis at SVR12, steatotic patients still had significantly higher VEGF levels. EGF was significantly increased in patients without steatosis but remained unchanged in patients with steatosis at SVR12. On the contrary, HGF levels significantly increased in patients with steatosis but not in non-steatotic patients, as shown in Figure 5.

## 4. Discussion

Here, we provide the evidence that CHC patients with SLD have distinct serum cytokine and growth factor profiles compared to patients without SLD during DAA treatment. This includes lower levels of Th2 cytokines (IL-4, IL-5, IL-9, IL-13), Th-9 (IL-9), Th-17 (IL-17A, IL-22) and anti-inflammatory IL-10, while there are no changes in Th2 or inflammatory IL-6 cytokine levels. On the contrary, SLD was linked with higher serum concentrations of EGF, VEGF and ANG. During CHC treatment, patients with SLD had distinct cytokine and growth factor kinetics, with SLD being the main source of variation in several cytokines (IL-5, IL-9, IL-10, IL-13, IL-17A, IL-22) and growth factors (EGF, VEGF and ANG), and it seems this was independent of fibrosis stage.

Surprisingly, there are only a few reports that analyze the characteristics of cytokine profiles in CHC depending on the presence of liver steatosis. At baseline, obesity and steatosis were associated with increased serum concentrations of CRP, IL-6 and TNFα [23,24]. Similarly, increased hepatic expression of IFN-γ, TNFα, IP-10 (CXCL10) and MCP-1 has been observed in steatotic patients with CHC [25,26].

While DAA treatment is associated with the improvement of liver inflammation and fibrosis regression in most patients, achieving sustained virological response might not completely restore HCV-induced impaired immune responses. Several reports showed only partial CD4+ T phenotype restoration, the irreversibility of monocyte activation and impaired IFN-γ and IL-2 cytokine production by HCV-specific CD4+ and CD8+ T cells upon HCV eradication [27,28]. HCV-specific CD8+ T cells remain functionally impaired despite HCV clearance, including the persistence of T-cell exhaustion, impaired cytokine production, proliferative capacity and mitochondrial and metabolic dysfunction [29]. While the majority of inflammatory mediators that increase before treatment significantly decrease in the plasma after HCV treatment, they do not reach levels comparable to healthy individuals [30]. The suppressed mediators (including IL-17, IL-1β, IFN-γ and IL-4) do not normalize even after 36 weeks of treatment initiation [30]. These studies indicated that impaired immune reconstitution might also be influenced by non-viral factors, such as age, sex or degree of liver fibrosis [6,27,28,30].

Notably, only one study examined the impact of steatosis on cytokine milieu after HCV eradication. Recently, Du Y et al. examined the association of 92 biomediators with steatosis grade in 94 patients with HCV genotype 1b infection [31]. The authors reported the correlation of SCF, TWEAK, FGF-21 and IL-18R1 with steatosis grade 96 weeks after virus clearance, while there was no difference in the concentrations of the other cytokines/interleukins measured [31]. Similarly, we did not find temporal changes in the majority of measured biomodulators during DAA treatment.

In this study, we focused on the role of IL-22, IL-9, IL-10 and IL-13 in CHC, particularly in the context of steatosis.

IL-10 plays a central role in homeostasis by balancing pro- and anti-inflammatory immune responses to ensure effective antimicrobial defenses while limiting immune system hyperactivation and potential tissue damage [32]. Elevated concentrations of IL-10 have been related to cytokine patterns favoring the establishment of chronic HCV infection, and experimental models of liver cirrhosis have demonstrated antifibrotic properties of this cytokine [33]. A recent study by Solleiro-Villavicencio et al. showed that concentrations of IL-10 in the liver tissue samples and sera of morbidly obese patients progressively decrease as lobular inflammation increases and proposed a hypothesis that the loss of IL-10-mediated anti-inflammatory counterbalance favors the development of lobular inflammation [34]. Similarly, the reduced concentrations of IL-10 in patients with steatosis observed in our study might indicate that the loss of anti-inflammatory properties of IL-10 significantly contributes to the establishment of steatosis and, possibly, the development of fibrosis.

IL-22, as a member of the IL-10 cytokine family, exhibits important tissue-protective effects by stimulating endothelial cell proliferation, migration and angiogenesis (for a review, see [32,35]. IL-22 exhibits a well-documented hepatoprotective effect that has been shown in several models of liver injury. Treatment with IL-22 ameliorated steatosis and liver damage by the upregulation of anti-apoptotic and anti-oxidative genes, and the vial downregulation of lipogenic genes in hepatocytes was mediated by activation of STAT3-signaling cascade [36]. In vitro and in vivo studies showed that the overexpression of IL-22 promoted liver regeneration, hepatocyte survival and proliferation [36]. Interestingly, our results have shown significantly lower serum concentrations of IL-22 in patients with steatosis, particularly in those with advanced liver fibrosis, as well as a negative correlation between CAP and serum levels of IL-22. These results probably reflect the importance of the IL-22 biological effects on hepatocytes in the livers of steatosis patients that subsequently reduced the levels of the soluble form of this cytokine.

IL-9 is a member of the γ_c_ cytokine family that shares a common cytokine receptor γ chain, which includes IL-2, IL-4, IL-7, IL-15 and IL-21 [37]. It is mainly synthesized by Th9 cells that, based on the microenvironment, can induce immune responses related to autoimmunity and inflammation or tolerance. Ali et al. have shown that viral clearance in hepatitis C is associated with a decrease in Th9 cells with significantly lower numbers of Th9 cells in patients who achieved SVR compared with treatment failures [38]. Reduced concentrations of IL-9 in the sera of patients with steatosis and a negative correlation between CAP and IL-9 levels shown in this study may suggest that reduced levels of IL-9 and Th9 cells may contribute to the microenvironment favoring the development of steatosis.

IL-13 is an important mediator of humoral immunity that is synthesized by Th2 helper cells. It stimulates the differentiation of B cells, inhibits the activation of macrophages, attenuates Th1 signaling in a variety of infectious and non-infectious diseases and is involved in the activation of fibroblasts. The role of IL-13 in liver-related diseases has been investigated in several models. Wong et al. analyzed a cohort of patients with chronic hepatitis B and found that the presence of metabolic syndrome and chronic inflammation were two independent factors leading to liver cirrhosis progression and that IL-13 played a central role in linking the inflammatory and metabolic processes [39]. Concentrations of IL-13 in the plasma were independently associated with increased CAP score in chronic hepatitis B patients, with every increase of 1 unit of IL-13 being associated with a 0.98 unit increase in CAP score [39]. Studies using the NAFLD model also highlighted the importance of IL-13 in the liver by showing that IL-13 was involved in progression from simple steatosis to metabolic-disfunction-associated steatohepatitis (for a review, see [40]). Our results have shown significantly decreased levels of IL-13 in patients with steatosis, particularly those exhibiting advanced liver fibrosis, highlighting the importance of this pleiotropic cytokine in the pathogenesis of CHC as well.

On the contrary, we found that patients with liver steatosis had significantly higher baseline concentrations of growth factors EGF, VEGF and ANG-2. Growth factors are important biological modulators that play a significant role in the pathogenesis of liver fibrosis and liver regeneration and possibly contribute to the development of HCC [41]. These were already analyzed in patients with liver diseases, but very few studies have examined their relationship with liver steatosis in patients with CHC.

Recently, higher levels of ANG-2 were demonstrated in CHC patients with NAFLD, which correlated with the degree of liver fibrosis [42]. It is well known that ANG-2 is an important factor in angiogenesis and a significant contributor to progression to hepatocellular carcinoma [43]. Liu et al. showed that the function of ANG-2 depends on its interactivity with other growth factors, and interaction between ANG-2 and VEGF can contribute to neo-vascularization [44]. Also, HCV assists VEGF through the stabilization of hypoxia-inducible factor 1 subunit α (HIF1-α), which increases angiogenesis [45].

Regarding EGF, Shehata et al. demonstrated a potential role of EGF in oncogenesis related to chronic viral hepatitis [46]. Virzi et al. demonstrated that the epithelial growth factor receptor (EGFR) induces mitogen-activated protein kinase (MAPK) signaling and the expression of genes related to fibrosis and hepatocyte proliferation [45].

When comparing the impact of liver fibrosis on patients with and without steatosis on serum growth factor concentrations, the serum level of HGF was increased in advanced fibrosis in patients with steatosis. Considering that pleiotropic cytokine HGF is involved in liver repair during fibrosis, these findings could indicate increased HGF serum levels due to liver regeneration after treatment with DAA [47]. Further, the degree of liver damage and hepatocellular dysfunction depend on the serum concentration of HGF [47], as recently highlighted by Marín-Serrano et al., who stated that the serum concentration of HGF is an unconventional biomarker related to the stage of fibrosis in CHC [48].

Finally, we sought to access the association of growth factor concentrations with steatosis and fibrosis regression after DAA treatment. When comparing patients with and without liver steatosis at SVR12, we found that there was a significant decrease in VEGF levels in patients with and without liver steatosis at SVR12, but steatotic patients still had significantly higher VEGF levels. Furthermore, EGF was significantly increased in patients without steatosis but remained unchanged in patients with steatosis at SVR12. On the contrary, HGF levels significantly increased in patients with steatosis but not in non-steatotic patients. These could be related to liver regeneration processes, since EGF, HGF and VEGF have been described as important regulators of liver regeneration and healing [49,50].

Our study should be viewed within its limitations. Since this was an observational study, causality could not be determined; fibrosis and steatosis were assessed by non-invasive technics, not by liver biopsy. The results cannot be generalized due to the limited study population, and the relatively small number of patients in genotypes 2 and 4 limits the statistical significance of the obtained results. For the same reason, the impact of the DAA regimen on biomarkers’ kinetics was not analyzed. Only early changes in cytokine and growth factor concentrations at selected timepoints during DAA treatment and at the SVR12 were analyzed, which might not reflect the late impact of liver steatosis on immune response restoration after HCV clearance.

Nevertheless, we studied a well-defined cohort of patients and reported the comprehensive data examining the interleukin and growth factor serum profiles in patients with liver steatosis and CHC. There is growing evidence that in chronic HCV infection, the milieu of soluble inflammatory mediators after viral clearance might be disrupted, and our study provided additional evidence that this might be driven by the presence of liver steatosis. This highlights the need to include liver steatosis as a variable when examining immune response in other clinical settings, such as viral hepatitis. Identifying the distinct cytokine profile in patients with SLD and CHC could have practical implications and could initiate new fibrosis management strategies. However, the long-term clinical impact of this immunological finding is yet to be defined.

## 5. Conclusions

In conclusion, we have shown that liver steatosis is associated with distinct cytokine and growth factor profiles and kinetics during CHC treatment, which might be associated with disease severity and the capacity for liver regeneration and contribute to fibrosis persistence in this patient group.

## Figures and Tables

**Figure 1 jcm-13-04849-f001:**
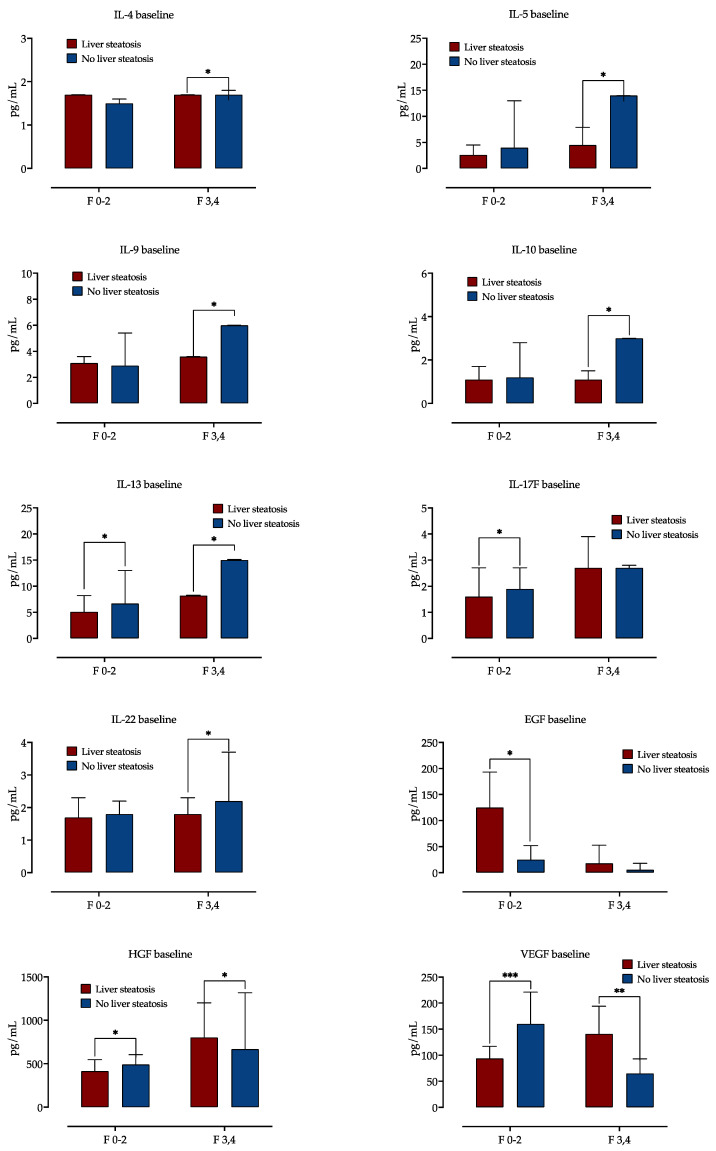
Baseline cytokine and growth factor serum concentrations according to presence of steatosis, stratified by fibrosis stage. Shown are medians with IQR, and difference significance is calculated by the Mann–Whitney test. *: *p* < 0.05; **: *p* < 0.01; ***: *p* < 0.001. Abbreviations: interleukine, IL; Epidermal growth factor, EGF; Hepatocyte growth factor, HGF; Vascular endothelial growth factor, VEGF.

**Figure 2 jcm-13-04849-f002:**
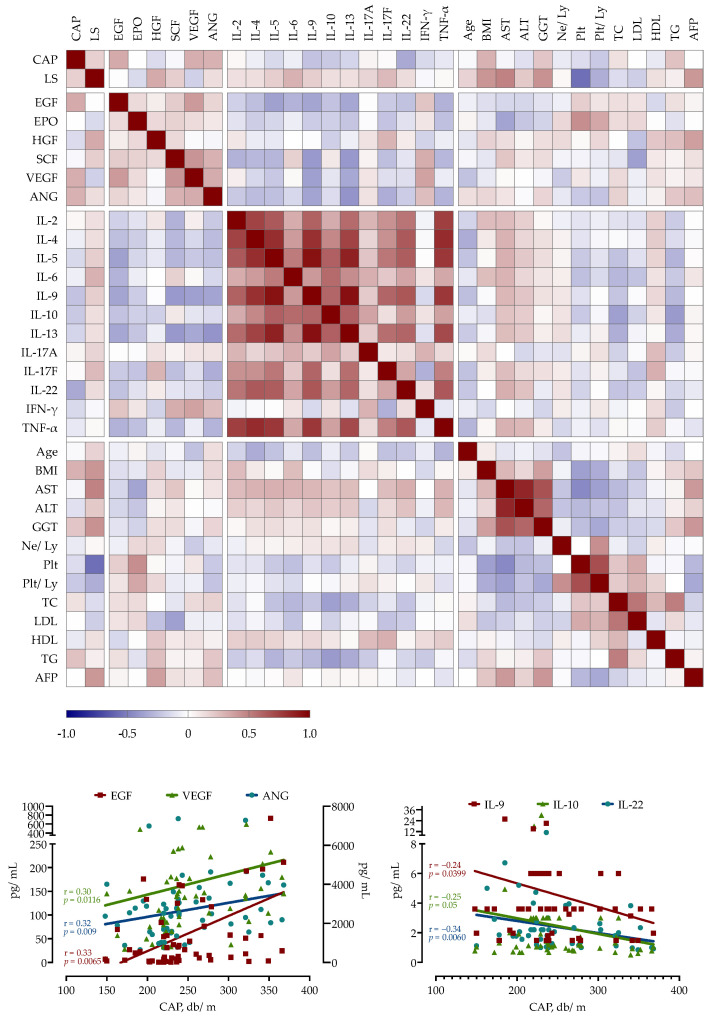
Spearman’s correlation correlogram. The strength of the correlation between two variables is represented by the color at the intersection of those variables. Colors range from dark blue (strong negative correlation; r = −1.0) to red (strong positive correlation; r = 1.0).

**Figure 3 jcm-13-04849-f003:**
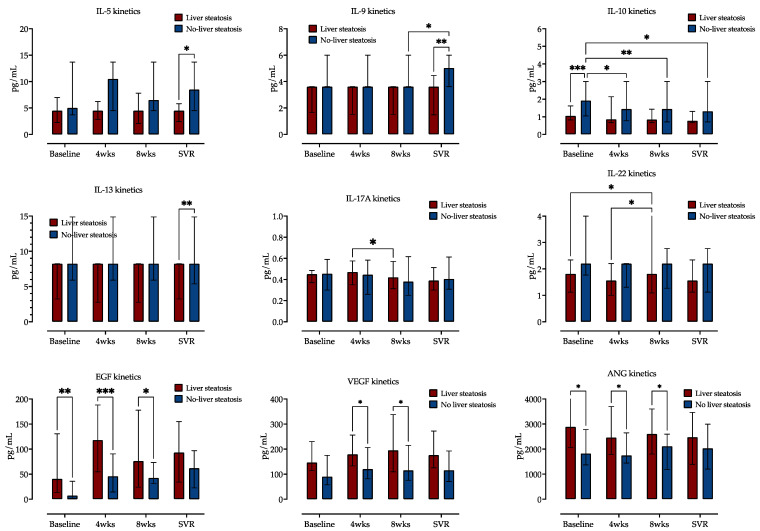
Comparison of cytokine and growth factor serum concentrations at four selected time points (before treatment, at weeks (wks) 4 and 8 of treatment and at time of sustained virological response, SVR), stratified by the presence of liver steatosis. Medians with IQRs are shown. Repeated measures two-way ANOVA with Tukey’s multiple comparisons test was used to calculate the source of the variations. *: *p* < 0.05; **: *p* < 0.01; ***: *p* < 0.001.

**Figure 4 jcm-13-04849-f004:**
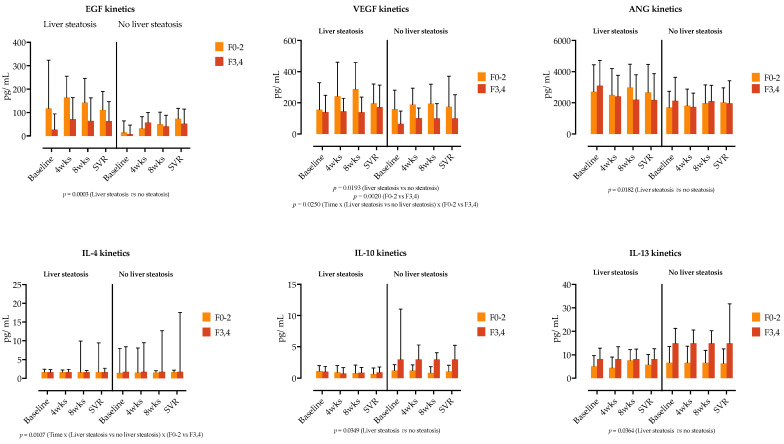
Comparison of growth factor and cytokine concentrations at four selected time points (before treatment, at weeks (wks) 4 and 8 of treatment and at time of sustained virological response. SVR), stratified by the presence of liver steatosis and liver fibrosis (F0–2 vs. F3,4). Medians with IQRs are shown. Repeated measures three-way ANOVA with Tukey’s multiple comparisons test was used to calculate the source of variations. *p*-values are shown; *p* < 0.05 is considered significant.

**Figure 5 jcm-13-04849-f005:**
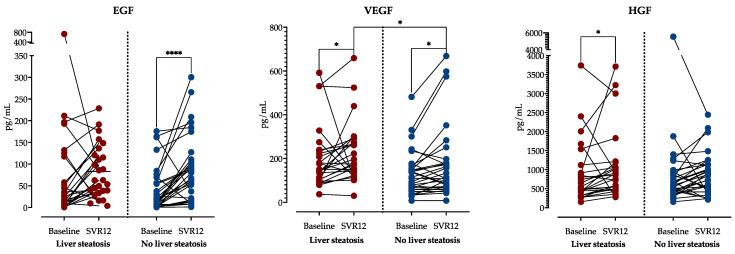
Comparison of growth factor concentrations before treatment and at SVR12 stratified by the presence of liver steatosis at SVR12. A Wilcoxon rank sum test was performed to analyze the differences in time within the groups, and the Mann–Whitney test analyzed the differences between the groups. *: *p* < 0.05; ****: *p* < 0.0001. Abbreviations: Epidermal growth factor, EGF; Hepatocyte growth factor, HGF; Vascular endothelial growth factor, VEGF; sustained virological response, SVR.

**Table 1 jcm-13-04849-t001:** Baseline patients’ characteristics.

	Liver Steatosis	No Liver Steatosis	*p*-Value *
Age, years	58 (45–63)	55 (45–63)	0.5995
Male sex	14 (50%)	18 (64.29%)	0.4182
Obesity (BMI > 30 kg/m^2^)	7 (25.0%)	1 (3.57%)	0.0511
BMI, kg/m^2^	26 (23–30)	25 (22–26)	0.0294
Duration of HCV infection, years	22 (14–30)	20 (10–32)	0.6505
HCV genotype			
1a	6 (21.4%)	6 (21.4%)	0.8009
1b	13 (46.4%)	16 (57.1%)	
3	7 (25.0%)	5 (17.9%)	
2	1 (3.6%)	1 (3.6%)	
4	1 (3.6%)	0	
HCV viremia, log10	5.7 (5.3–6.3)	6.0 (5.7–6.3)	0.1389
Liver stiffness, kPa	14 (9.5–20)	12 (6.9–19)	0.2319
CAP, dB/m	277 (244–324)	219 (192–224)	<0.0001
Advanced fibrosis (F3–4)	17 (60.71%)	16 (57.14%)	0.9999
DAA treatment regiment			
Sofosbuvir + velpatasvir	13 (46.4%)	10 (35.7%)	0.7157
Glecaprevir + pibrentasvir	11 (39.3%)	13 (46.4%)	
Elbasvir + grazoprevir	4 (14.3%)	5 (17.9%)	
**Comorbidities**			
T2DM	6 (21.43%)	2 (7.14%)	0.2516
Arterial hypertension	13 (46.43%)	8 (28.57%)	0.2695
Dyslipidemia	6 (21.43%)	3 (10.71%)	0.4688
Cardiovascular comorbidities	2 (7.14%)	1 (3.57%)	0.9999
PWID	7 (25.0%)	4 (14.29%)	0.5027
**Laboratory**			
Hemoglobin	144 (132–157)	139 (129–149)	0.3423
Leukocytes, ×10^9^	6.2 (5.1–6.8)	6.3 (4.7–7.1)	0.7112
Neutrophil-to-lymphocyte ratio	1.5 (1.0–2.0)	1.7 (1.3–2.5)	0.1221
Platelets	152 (117–188)	190 (134–217)	0.1231
Platelets-to-lymphocyte ratio	78 (53–102)	96 (70–123)	0.0497
Total bilirubin	14 (11–20)	11 (10–17)	0.1353
AST	55 (40–92)	58 (34–91)	0.8997
ALT	56 (37–93)	64 (40–32)	0.7791
GGT	56 (38–120)	40 (32–75)	0.1230
ALP	86 (65–105)	81 (69–107)	0.7604
AFP	4.7 (2.5–9.2)	3.7 (2.4–9.4)	0.7235
Albumins, g/L	42 (40–44)	43 (41–46)	0.2381
Fasting glucose	6.1 (5.3–7.8)	5.5 (4.7–6.1)	0.0312
Cholesterol	4.8 (4.2–5.3)	4.3 (3.2–5.4)	0.2065
LDL	2.5 (2.0–2.8)	2.6 (1.8–3.0)	0.9500
HDL	1.3 (1.1–1.7)	1.4 (1.0–1.7)	0.7904
Triglycerides	1.7 (1.1–2.2)	1.3 (0.83–1.9)	0.1861

* Fisher’s exact or the Mann–Whitney U test, as appropriate.

**Table 2 jcm-13-04849-t002:** Baseline patients’ cytokine and growth factor concentrations according to steatosis grade.

pg/mL	Liver Steatosis	No Liver Steatosis	*p*-Value *
IL-2	0.73 (0.61–0.77)	0.73 (0.63–0.73)	0.7824
IL-4	1.7 (0.88–1.7)	1.7 (1.2–1.7)	0.3473
IL-5	4.5 (2.3–7.0)	5.0 (3.7–14)	0.1154
IL-6	6.2 (6.0–9.5)	7.4 (6.2–14)	0.2426
IL-9	3.6 (1.7–3.6)	3.6 (3.6–6.0)	0.0227
IL-10	1.1 (0.82–1.6)	1.9 (1.1–3.0)	0.0188
IL-13	8.2 (3.2–8.2)	8.2 (5.9–15)	0.0445
IL-17A	0.45 (0.37–0.49)	0.46 (0.30–0.59)	0.7850
IL-17F	2.6 (1.4–2.7)	2.3 (2.0–2.7)	0.7059
IL-22	1.8 (1.1–2.3)	2.2 (1.8–4.0)	0.0201
TNF-α	4.5 (1.5–6.2)	5.6 (4.5–6.0)	0.1958
IFN-γ	9.8 (5.7–10)	9.8 (5.0–17)	0.3423
**Growth factors**			
EGF	40 (13–130)	7.2 (2.0–36)	0.0017
EPO	398 (293–634)	418 (274–598)	0.8997
HGF	526 (348–901)	597 (439–939)	0.4663
SCF	102 (67–117)	82 (52–155)	0.9385
VEGF	146 (114–230)	90 (57–175)	0.0102
ANG	2884 (2063–4074)	1822 (1368–2781)	0.0110

* A *t*-test or Mann–Whitney U test, as appropriate

## Data Availability

The datasets generated during and/or analyzed during the current study are available from the corresponding author upon reasonable request.

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
