# Peer review of "The Impact of Liver Steatosis on Interleukin and Growth Factors Kinetics during Chronic Hepatitis C Treatment"

_jcm, 2024, doi:10.3390/jcm13164849_

Round 1

Reviewer 1 Report

Comments and Suggestions for Authors

This is a research paper on the effects of hepatic lipidosis on interleukin and growth factor 2 kinetics during chronic hepatitis C treatment. The conclusions are interesting and promising. However, there are some criticisms.

1) Are there any patients in this cohort who have undergone liver biopsy, and I suspect that CAP values alone may not be sufficient to rule out steatohepatitis. 

2) There was no mention of a DAA regimen. It would be interesting to know if there are differences, if any, due to this and I would appreciate your consideration. 

Author Response

This is a research paper on the effects of hepatic lipidosis on interleukin and growth factor 2 kinetics during chronic hepatitis C treatment. The conclusions are interesting and promising. However, there are some criticisms.

COMMENT 1: Are there any patients in this cohort who have undergone liver biopsy, and I suspect that CAP values alone may not be sufficient to rule out steatohepatitis. 

RESPONSE 1: We agree with the reviewer that liver biopsy is the best way to examine steatohepatitis in patients with CHC. None of the patients had liver biopsy and this is important limitation. This is now highlighted in the limitations section.  

COMMENT 2: There was no mention of a DAA regimen. It would be interesting to know if there are differences, if any, due to this and I would appreciate your consideration. 

RESPONSE 2: We have added DAA regiment in the Results section (Table 1, Baseline patients’ characteristics). We agree that hypothetically DAA regiment might influence cytokine kinetics and development/persistance of steatosis and this should be examined in a larger study. In our study additional subgroup analysis based on DAA regiment could not be performed due to the small number  of patients in each subgroup (steatosis +/-; fibrosis +/-, DAA type) that limits statistical anaysis. This is now added in limitations.

Reviewer 2 Report

Comments and Suggestions for Authors

I have the pleasure to read ana analyze the manuscript "The impact of liver steatosis on interleukin and growth factors kinetics during chronic hepatitis C treatment" by Matotek et al. The paper is well written, presenting the results of a study analyzing the cytokines level in patients with chronic hepatitis C. The authors aimed to investigate the interleukines and growth factor profile in patients with chronic hepatitis C during DAA treatment with or without steatosis. 

The study was well organized and presented. However, the presentation of the results could be improved. There are a lot of data and figures, and some of them seem too small. Data from Figures 3, 4, and 5 may be presented in a table and/ or enlarged images (Figure 5).

The results' analysis is presented in detail in the Discussions section. The authors also show their study's limitations. Still, I think they must emphasize the relevance of the results for clinical practice and the possibility of further research based on the present results.

Comments on the Quality of English Language

The use of the English language is good. There are some minor adjustments and editing (use of abbreviations; include the abbreviation explanation in the Legend of the tables).

Author Response

I have the pleasure to read and analyze the manuscript "The impact of liver steatosis on interleukin and growth factors kinetics during chronic hepatitis C treatment" by Matotek et al. The paper is well written, presenting the results of a study analyzing the cytokines level in patients with chronic hepatitis C. The authors aimed to investigate the interleukines and growth factor profile in patients with chronic hepatitis C during DAA treatment with or without steatosis. 

COMMENT 1: The study was well organized and presented. However, the presentation of the results could be improved. There are a lot of data and figures, and some of them seem too small. Data from Figures 3, 4, and 5 may be presented in a table and/ or enlarged images (Figure 5).

RESPONSE 1: We thank the reviewer for this comment. Figures 3 and 4 are now presented in Supplementary Materials as enlarged images. Figure 5 is now enlarged.

COMMENT 2: The results' analysis is presented in detail in the Discussions section. The authors also show their study's limitations. Still, I think they must emphasize the relevance of the results for clinical practice and the possibility of further research based on the present results.

RESPONSE 2: Thank you for your comment. We have added brief paragraph to emphasise the relevance of our results at the end of the discussion.